# The Impact of Dietary Flavonols on Central Obesity Parameters in Polish Adults

**DOI:** 10.3390/nu14235051

**Published:** 2022-11-27

**Authors:** Joanna Popiolek-Kalisz

**Affiliations:** 1Clinical Dietetics Unit, Department of Bioanalytics, Medical University of Lublin, ul. Chodzki 7, 20-093 Lublin, Poland; joannapopiolekkalisz@umlub.pl; 2Department of Cardiology, Cardinal Wyszynski Hospital in Lublin, al. Krasnicka 100, 20-718 Lublin, Poland

**Keywords:** obesity, abdominal obesity, flavonols, quercetin, myricetin, isorhamnetin

## Abstract

Background: Central obesity is defined as the excessive fat tissue located in abdominal region accompanied by systemic inflammation, which drives to cardiovascular disease. Flavonols are antioxidative agents present in food. The aim of this study was investigating the relationship between dietary flavonols intake and central obesity. Methods and results: 80 participants (40 central obese and 40 healthy controls) were administered a food frequency questionnaire dedicated to flavonols intake assessment. Body composition was measured with bioelectrical impedance analysis. The analysis showed significant differences between central obese participants and healthy controls in total flavonol (*p* = 0.005), quercetin (*p* = 0.003), kaempferol (*p* = 0.04) and isorhamnetin (*p* < 0.001) habitual intake. Among central obese participants, there was a moderate inverse correlation between fat mass (FM) and total flavonol (R = −0.378; 95% CI: −0.620 to −0.071; *p* = 0.02), quercetin (R = −0.352; 95% CI: −0.601 to −0.041; *p* = 0.03), kaempferol (R = −0.425; 95% CI: −0.653 to −0.127; *p* = 0.01) and myricetin intake (R = −0.352; 95% CI: −0.601 to −0.041; *p* = 0.03). BMI was inversely correlated with total flavonol (R = −0.330; 95% CI: −0.584 to −0.016; *p* = 0.04) and quercetin intake (R = −0.336; 95% CI: −0.589 to −0.023; *p* = 0.04). Waist circumference was inversely correlated with total flavonol (R = −0.328; 95% CI: −0.586 to −0.009; *p* = 0.04), quercetin (R = −0.322; 95% CI: −0.582 to −0.002; *p* = 0.048) and myricetin intake (R = −0.367; 95% CI: −0.615 to −0.054; *p* = 0.02). Among flavonols’ dietary sources, there was an inverse correlation between black tea consumption and FM (R: −0.511; 95% CI: −0.712 to −0.233; *p* < 0.001) and between coffee and waist circumference (R: −0.352; 95% CI: −0.604 to −0.036; *p* = 0.03) in central obese participants. Conclusions: The higher flavonol intake could play a protective role in abdominal obesity development. What is more, total and selected flavonol dietary intakes are inversely correlated with the parameters used for obesity assessment in central obese participants. The habitual consumption of products rich in flavonols, mainly tea and coffee, could possibly have a preventive role in abdominal obesity development.

## 1. Introduction

Civilizational diseases are the major cause of deaths worldwide, including ischemic heart disease responsible for 16% of world’s total deaths, stroke responsible for 11% and diabetes, which followed a 70% increase from 2000 [1]. All these conditions are related to their risk factors, which involve elevated blood pressure, hyperglycemia and dyslipidemia, while central obesity is one of the causes that can lead to all of the mentioned conditions. What is more, it is proven that elevated body mass index (BMI) and waist circumference are strongly associated with, for example, cardiovascular disease (CVD) morbidity [2]. Higher values of BMI are also associated with higher mortality in CVD [3]. Throughout last years, there has been observed a clear trend toward obesity burden [4,5]. Thus, proper obesity controlling is one of the strategies not only for civilizational diseases prevention, but also for further management [6].

Obesity is the result of a misbalance between energy intake and increase in the triglycerides which are stored in adipocytes. In everyday practice, BMI is a low-cost parameter which enables easy obesity detection and classification. BMI is calculated by dividing body mass (in kilograms) by square of height (in meters). According to the World Health Organization, obesity is diagnosed when BMI is 30 kg/m^2^ or higher. Then, obesity classes are also categorized on the basis of BMI value. Class 1 means BMI from 30 kg/m^2^ to 34.99 kg/m^2^, class 2 is BMI from 35 kg/m^2^ to 39.99 kg/m^2^ and class 3 is BMI 40 kg/m^2^ or higher. On the other hand, the American Association of Clinical Endocrinologists and American College of Endocrinology guidelines propose different approach combining the application of BMI, waist circumference and adiposity measurements with available methods [7]. These guidelines underline that BMI alone cannot identify excess adiposity and serve for obesity diagnosis in all cases [7,8]. Fat distribution plays a role in metabolic consequences, as abdominal obesity is associated with higher risk of insulin resistance (IR), hypertension and dyslipidemia. Central obesity in Europe is clinically defined as a waist circumference ≥94 cm for men and ≥80 cm for women, while in other regions, the cut-off points may differ, e.g., >86 cm for women and >102 cm for men in the United States [7,9].

Bioelectrical impedance analysis (BIA) is a noninvasive method of body composition assessment. It is based on passing alternating electrical current through the body. Different body tissues present different electrical properties, thus the changes of impedance (Z), which is a combination of resistance (R) and reactance (Xc), can serve as direct bioelectrical parameter or can be transmuted into information about selected tissues content. This way BIA gives information about body fat mass (FM), total body water (TBW), extracellular water (ECW), intracellular water (ICW) and fat-free mass (FFM). The measurement can be taken at different electric current frequencies, usually at 5, 50, 100 or 200 kHz. BIA parameters can be used for general body composition analysis and for nutritional status assessment [10,11].

Excessive FM, particularly in the form of central obesity is associated with an inflammatory response which contributes to endothelium disfunction and atherosclerosis development. Adipose tissue inflammation, called *adipositis*, involves the production of cytokines and adipokines, such as tumor necrosis factor α, interleukin 1β and interleukin 6 [12,13,14]. These mechanisms are strongly presented in visceral fat tissue, as it contains more macrophages comparing to fat tissue in other regions [15]. It leads to numerous metabolic complications of obesity, including IR, which results in diabetes and dyslipidemia, whichthen results in atherosclerosis. Counteracting these conditions could be performed by targeting their primary mechanisms, including inflammation. Therefore, searching for agents with anti-inflammatory potential is one of the leading directions of dietary approach against obesity and its complications [16]. Flavonols are a group of flavonoids, and present antioxidative properties. The most abundant individual flavonol is quercetin, followed by kaempferol, myricetin, and isorhamnetin [17,18,19]. Nonetheless, this group includes also less prevalent ones, which are morin, galangin, fisetin, kaempferide, azaleatin, natsudaidain, pachypodol and rhamnazin [20]. The major products that contribute to dietary flavonols intake are onions [21], tea [22], apples [23], kale [24], lettuce [25], tomatoes [26], broccoli [27], grape skins [28] which are responsible for the flavonol contents in red wine [29], citrus fruits [30,31], and berries [19,20,32,33].

There are several papers that showed the positive impact of flavonols on obesity parameters; however there are not many of them which analyze the role of specific compounds [34,35]. This information would be helpful not only to indicate the direction of future studies regarding the dietary model, but also potential interventional studies and supplementation, which require investigating a single compound. What is more, there are not available any papers investigating the role of kaempferol, myricetin and isorhamnetin in obesity prevention in humans.

Therefore, the aim of this paper was to investigate the differences in dietary flavonol intake between participants with central obesity and healthy controls. Then the relationship between habitual intake of total and selected flavonols (quercetin, kaempferol, myricetin and isorhamnetin) and their sources and parameters used for obesity assessment and body composition in central obese patients was examined.

## 2. Materials and Methods

### 2.1. Study Group

In total, 80 participants (40 central obesity patients and 40 healthy controls) were enrolled in the study between March and November 2022. The inclusion criteria for the study group were as follows: (1) waist circumference ≥ 94 cm for men and ≥80 cm for women, (2) age 18–85 years, (3) written consent, (4) mental condition that enabled one-year retrospective dietary interview (i.e., answering the question: “How often during the last year did you consume the suggested portion of the following product?”), and (5) lack of conditions that interfere with BIA measurement (metal implants, e.g., orthopedical or pacemakers, abnormal body geometry, e.g., after amputations, pregnancy, and abnormal body temperature). The exclusion criteria were (1) lack of written consent, (2) abnormal mental condition, (3) presence of metal implants, (4) pregnancy, (5) amputations, (6) waist circumference < 94 cm for men and <80 cm for women, (7) age below 18 years or above 85 years, and (8) special diet due to health reasons.

Healthy controls were volunteers, who matched the following criteria: (1) waist circumference < 94 cm for men and <80 cm for women, (2) age 18–85 years, (3) written consent, (4) mental condition that enabled one-year retrospective dietary interview, and (5) lack of any chronic disease or chronic treatment. The exclusion criteria for the control group were (1) lack of written consent, (2) abnormal mental condition, (3) pregnancy, (4) special diet due to health reasons, (5) waist circumference ≥ 94 cm for men and ≥80 cm for women, (6) age below 18 years or above 85 years, and (7) chronic disease or treatment. All the participants were non-smokers.

### 2.2. Methods

#### 2.2.1. Flavonol Intake Assessment

The food frequency questionnaire dedicated for specific flavonol long-term (one year) intake assessment was administered to the participants [36]. The questionnaire provided information upon the mean consumption of 140 products which are sources of flavonols during a preceding year. The suggested portions of the products were introduced on the basis of typical servings in everyday life (e.g., one piece, a glass) and described by both—a suggested serving (e.g., a piece, a glass) and a weight in grams. Then, on the basis of the declared frequency of selected products consumption (never or almost never, once a month, few times a month with a number of times per month given by the responder, once a week, few times a week with a number of times per week given by the responder, once a day, few times every day with a number of times per day given by the responder), the mean daily consumption of each product was calculated. The amounts of quercetin, kaempferol, isorhamnetin and myricetin in each product were based on the data available in the USDA database [37]. On the basis of it, the mean daily intake of quercetin, kaempferol, isorhamnetin, and myricetin was calculated for each participant. Total flavonol intake was calculated by adding the values for quercetin, kaempferol, isorhamnetin and myricetin. Dietary intakes were then calculated for body mass ratio by dividing the result of the mean daily intake of each compound by the measured body mass.

#### 2.2.2. Anthropometrical and Body Composition Measurements

The anthropometrical measurements included body mass, height and waist circumference. The participants’ body mass was measured with 0.05 kg accuracy with WTL-150A scale (Lubelskie Fabryki Wag, Lublin, Poland) by the trained professional. The participants were permitted to wear only underwear for this measurement. Waist circumference was measured with measuring tape with 0.5 cm accuracy in the mid-horizontal plane between the superior iliac crest and the lower margin of the last rib and height with stadiometer with 1.0 cm accuracy by the trained professional [38].

BIA measurement was performed with ImpediMed SFB7 (Impedimed Ltd., Brisbane, Australia) in accordance with the producer’s instructions. The system was based on four self-adhesive surface electrodes connected to the analyzer. Electrodes were placed on the washed and dried skin of the left wrist and left ankle. The measurements were performed in the horizontal position with limbs resting loosely at 30–45 degrees to the body. Before the examination, participants had to lie in position for 5 min and were not allowed to drink, eat, or exert any physical effort in the preceding three hours. The measurements were taken in triplicate, and then the mean values were used for statistical analysis.

#### 2.2.3. Statistical Analysis

Statistical analyses were performed with the RStudio software v. 4.2.0. The normality of the distribution of each parameter was checked by the Shapiro–Wilk test. The variables were presented as means (±SD). The Mann–Whitney test was used to compare the flavonol intakes between obese participants and healthy controls. *p* value below 0.05 was considered significant. The Pearson correlation was used to analyze the association between selected flavonol mean daily intake and BIA parameters, waist circumference or BMI. The cut-off points used for correlation coefficient were as follows: ≤0.29 as low, 0.30–0.49 as moderate and ≥0.50 as high correlation. Confidence interval (CI) level was 95%.

#### 2.2.4. Ethics

The study was approved by the local Bioethics Committee of the Medical University of Lublin (consent no. KE-0254/9/01/2022). The study was conducted in line with the directives of the Declaration of Helsinki on Ethical Principles for Medical Research. All participants signed a written consent.

## 3. Results

### 3.1. General Characteristics of the Study Population

The final study group included 39 central obese participants (20 men and 19 women) and 40 healthy controls (11 men and 29 women). One obese participant was excluded from the analysis due to filling in the questionnaire incompletely. The mean body mass of central obese participants was 80.18 ± 10.35 kg, BMI was 29.18 ± 3.75 kg/m^2^, and waist circumference was 102.72 ± 11.09 cm. Anthropometrical and body composition characteristics of the central obese group are presented in Table 1. The mean body mass of the healthy control group was 59.48 ± 10.43 kg, BMI was 20.94 ± 1.65 kg/m^2^ and waist circumference was 65.6 ± 6.55 cm. The comparison of the anthropometrical characteristics of the central obesity and control group is presented in Table 2.

### 3.2. Central Obesity and Healthy Participants Comparison

The analysis revealed significant differences in most of the flavonol intake between central obesity and healthy participants. The central obese participants presented significantly lower mean daily total flavonols intake (0.82 ± 0.44 mg/kg vs. 1.18 ± 0.66 mg/kg; *p* = 0.005), quercetin (0.52 ± 0.29 mg/kg vs. 0.77 ± 0.45 mg/kg; *p* = 0.003), kaempferol (0.19 ± 0.11 mg/kg vs. 0.25 ± 0.15 mg/kg; *p* = 0.04) and isorhamnetin (0.03 ± 0.02 mg/kg vs. 0.08 ± 0.07) mg/kg; *p* < 0.001. The detailed results are presented in Table 3. The boxplots showing these results are presented in Figure 1.

### 3.3. Body Composition in Central Obese Participants

The analysis showed significant moderate inverse correlation between quercetin intake and FM (R = −0.352; 95% CI: −0.601 to −0.041; *p* = 0.03). Kaempferol intake negatively correlated with TBW percentage (R = 0.330; 95% CI: 0.016 to 0.585; *p* = 0.04), FFM percentage (R = 0.321; 95% CI: 0.006 to 0.578; *p* = 0.046), FM (R = −0.425; 95% CI: −0.653 to −0.127; *p* = 0.01) and FM percentage (R = −0.330; 95% CI: −0.585 to −0.016; *p* = 0.04). Myricetin intake inversely correlated with FM (R = −0.352; 95% CI: −0.601 to −0.041; *p* = 0.03). Total flavonol intake correlated with FM (R = −0.378; 95% CI: −0.620 to −0.071; *p* = 0.02). Detailed results are presented in Table 4.

### 3.4. Anthropometrical Parameters in Central Obese Participants

The analysis showed a significant moderate inverse correlation between quercetin intake and BMI (R = −0.336; 95% CI: −0.589 to −0.023; *p* = 0.04) and waist circumference (R = −0.322; 95% CI: −0.582 to −0.002; *p* = 0.048). Myricetin intake inversely correlated with waist circumference (R = −0.367; 95% CI: −0.615 to −0.054; *p* = 0.02). Total flavonol intake correlated negatively with BMI (R = −0.330; 95% CI: −0.584 to −0.016; *p* = 0.04) and waist circumference (R = −0.328; 95% CI: −0.586 to −0.009; *p* = 0.04). Detailed results are presented in Table 4.

### 3.5. Flavonols’ Sources in Central Obese Participants

The major contributors to flavonol intake were onions (white and red), tomatoes, blueberries, apples, tea (black and green), coffee and wine. The analysis on the relationship between their consumption and mentioned above parameters used for obesity assessment (FM, FM percentage, waist circumference and BMI) revealed significant strong correlation between black tea consumption and FM (R: −0.511 95% CI: −0.712 to −0.233; *p* < 0.001) or FM% (R: −0.522; 95% CI: −0.719 to −0.247; *p* < 0.001). Waist circumference was significantly correlated with tomatoes (R: 0.370; 95% CI: 0.057 to 0.617; *p* = 0.02) and coffee consumption (R: −0.352; 95% CI: −0.604 to −0.036; *p* = 0.03). Detailed results are presented in Table 5.

## 4. Discussion

Obesity, particularly abdominal obesity, is one of the conditions leading to the development of civilizational diseases, which is why obesity prevention could potentially modify their progress [39]. Obesity is most often the result of excessive caloric intake; however, qualitative differences could also play a potential role in obesity development. Depending on the used parameter, obesity is defined as BMI ≥ 30 kg/m^2^, while abdominal obesity is diagnosed for waist circumference ≥ 94 cm for men and ≥80 cm for women in Europe [9]. Even though obesity is described as the presence of excessive fat tissue, there is not any consensus for cut-off points for FM or FM percentage to diagnose obesity [40]. Obesity is one of the most common health problems. In Poland, the central obesity prevalence is 45.7% for women and 32.2% for men in the total population [41].

Systemic inflammation, which accompanies obesity, leads to endothelium dysfunction and promotes atherosclerosis progression. Central obesity, which is the excessive fat tissue in the abdominal area, is a particular risk factor of diseases such as CVD or diabetes [42]. As obesity is a condition accompanied by the chronic low-grade inflammation of adipose tissue, the potential role of antioxidative agents has been emerging in recent years [43,44]. That is why flavonols’ impact on civilizational diseases and their risk factors was suggested in numerous studies [45,46,47]; however, its relationship with central obesity has not been established yet. It is an important point of view, as central obesity is one of the CVD and diabetes risk factors. What is more, there are no available studies conducted in humans that have examined the relationship between flavonols other than quercetin and central obesity. This study analyzed the correlation between myricetin, isorhamnetin and kaempferol intake and central obesity. The presented study also showed that there is an association between selected flavonol intake and the anthropometrical and body composition parameters is central obese patients.

This study showed the significant differences in flavonol habitual intakes between participants with central obesity and healthy controls. This relationship was observed for total flavonols and for selected compounds, such as quercetin, kaempferol and isorhamnetin. What is more, among the central obese participants, the general flavonol intake was also moderately inversely correlated with FM, BMI and waist circumference. Thus, we can provide a suggestion that the habitual consumption of products rich in flavonols could possibly have a preventive role in central obesity development. This observation is consistent with the results from the Korean population, in which dietary flavonol intake was inversely correlated with abdominal obesity prevalence in women [34]. However, in the same study, flavonol intake was also positively correlated with obesity based on BMI in men [34]. This difference might be caused by the fact that the authors of the mentioned study used the absolute flavonol intake instead of mean intake related to, for example, body mass. What is more, there are methodological differences between the study of Kim et al. and the presented research, as the study in the Korean group was based on 24 h recall and dual X-ray absorptiometry, while in the presented study, the intake of flavonols was investigated in one-year perspective, and body composition was assessed with BIA, which is more available in clinical background [34]. Moreover, these both studies are observational types, and they need interventional continuation to support their results.

Flavonols are a group of flavonoids that share a 3-hydroxyflavone backbone; nonetheless, the single compounds differ, e.g., by the presence and position of hydroxyl groups. These structural differences impact the bioactivity of these compounds [48]. Quercetin is more reactive than kaempferol due to the presence of an additional hydroxyl group at the R1 position [48]. This could be one of the potential reasons for the diverse impact strength on obesity parameters of different flavonols in central obese participants. What is more, the heat treatments, storage systems of the foods or cultivation practices could also impact the flavonols’ bioavailability and bioactivity. It was shown that cooking spinach leads to quercetin and kaempferol residues’ reduction [49]. A similar observation was made in berries, as cooking lead to quercetin loss—minor in strawberries and significant in bilberries [50]. On the other hand, the heat treatment of blueberries [51] or white beans [52] did not change their quercetin and kaempferol content, while in wine pomace, it even increased the flavonol content [53]. Storage could also impact flavonol content in some cases, as after 9 months of storage, quercetin loss was observed in bilberries and lingonberries, but not in black currants or red raspberries [50]. What is more, kaempferol and myricetin were even more susceptible to losses during storage [50]. Cultivation conditions could also impact phenolic content of foods, as higher UV exposition leads to higher quercetin content in red lettuce [54]. The differences in flavonol content could be also affected by the cultivar. These variabilities were described, for example, in sea buckthorn [55], plums [56] or grapes [29].

Quercetin is the most widespread individual flavonol in the everyday diet, as it is responsible for most of the observed relationships. It is present mainly in tea [22], onions [21] and berries (with whortleberry, lingonberry and cranberry being the most rich ones) [32]. Quercetin intake was inversely correlated with FM, BMI and waist circumference in central obese participants. Although the study by Nishimura et al. showed that quercetin supplementation (60 mg/day) did not change the amount of abdominal fat in general healthy group, the authors observed a reduction in the visceral fat amount in participants with lower levels of high density cholesterol (HDL-C) [35]. When high doses of quercetin were supplemented (500 or 1000 mg), they did not have impact on body mass or composition in healthy individuals [57]. On the other hand, quercetin supplementation was reported as being beneficial in term of hormonal balance in overweight and obese women [58]. It also reduced blood pressure and improved the lipid profile in overweight patients at risk of CVD [45,59]; however, the strength of its impact could also depend on genotype [60]. This may suggest that the quercetin impact on selected parameters could be present mainly in patients with excessive body mass or in combination with other flavonols, as they also presented potential impact on obesity parameters.

Kaempferol, the second prevalent flavonol, which is present mainly in kale [61] and spinach [49], intake was moderately inversely correlated with FM and FM percentage in central obese participants. Even though its role does not seem as critical as quercetin in terms of central obesity prevention (*p* = 0.003 for quercetin vs. *p* = 0.04 for kaempferol), they could possibly co-operate in their clinical effects. This corresponds with the results from animal model studies [62,63]. This study is the first one to link kaempferol to central obesity prevention in humans.

Isorhamnetin intake, whose main dietary contributor is onion [64], differed between central obese participants and healthy control, so we might assume that it could present protective properties against central obesity development; however, this is the first observation upon this topic in humans ever. On the other hand, once the central obesity condition is present, isorhamnetin intake was not correlated with the measured parameters. This observation is in opposition to the results presented by Rodriguez-Rodriguez et al. who observed metabolic benefits of isorhamnetin supplementation in obese mice [65]. These differences might be caused by the fact the in the mentioned study, the authors analyzed the metabolic changes only at the cellular level, such as glucose transporter 2 and peroxisome proliferator-activated receptor gamma mRNA content. The other reason could be the animal model used for the mentioned study. There are no available studies which focus on isorhamnetin and central obesity occurrence or parameters in humans, so more of them are definitely needed to support these observations.

Myricetin, which is less present in the everyday diet, but present mainly due to coffee consumption [64], intake was moderately inversely correlated with FM and waist circumference in central obese participants. Nonetheless, it was the only individual flavonol whose consumption did not differ significantly between central obese participants and the healthy control; thus, we can assume that its effects refer more to progression inhibition than to primary protection against central obesity. This observation is also in alignment with the results from animal model studies [66]. Nonetheless, this is also the first study that proved the potential relationship between obesity parameters and myricetin intake in humans.

The dietary sources of flavonols usually contain a set of them instead of one separated compound. As mentioned above, the co-operation between different individual flavonols could possibly also interfere with their biological effects and affect the strength of the impact on obesity parameters. That is why the relationship between the main dietary sources of flavonols and obesity parameters was also investigated. It showed that black tea consumption was strongly inversely correlated with FM and FM percentage. As tea is one of the main sources of quercetin, this observation is in line with the results referring to individual compounds and with animal model studies [67]. Numerous studies proved the antioxidative properties of tea polyphenols [68,69,70]. Nonetheless, the mechanism of tea impact on obesity development is complex, and aside from antioxidative potential, it included, for example, microbiome and nutrient intake interactions or protein kinase activation [67,71]. Black and green tea polyphenols also impact lipid metabolism from the level of intestinal digestion, e.g., emulsification to lipid accumulation, as black tea polyphenols can activate AMP-kinase involved in lipid metabolism or act on the nuclear receptors [72,73,74]. The majority of studies which suggest the beneficial role of tea for obesity focused on green tea [75]. Green tea is non-fermented in opposition to fermented black tea, which is why green tea presents higher antioxidative potential [76]. It is interesting that green tea consumption, which is even more rich in flavonols, did not present such a relationship with the obesity parameters in this study. It could be possibly the result of the much lower local consumption of green tea compared to black tea (0.48 portions/day vs. 1.81 portions/day). On the other hand, black tea polyphenols present some properties crucial in obesity development at a higher level compared to green tea, e.g., the inhibition of lipid emulsification during digestion [73].

As mentioned above, it is worth noting that mean tea consumption and quercetin intake did present different impact strength on obesity parameters, as quercetin intake was moderately inversely correlated with BMI, waist circumference and FM, while black tea consumption was strongly inversely correlated with FM and FM percentage. It could be the results of the fact that, even though black tea is one of the major quercetin sources, it contains also other bioactive compounds, such as catechins or gallocatechins, which also present these properties [77,78,79]. That is why the strength of this impact on FM could be potentially a result of the co-operation between quercetin and other compounds.

The preventive role of coffee consumption on central obesity is also suggested on the basis of the presented results. Mean coffee intake was moderately inversely associated with waist circumference, which is consistent with the results of the meta-analysis by Lee et al. [80]. Coffee is an important source of flavonols; however, it contains also other bioactive compounds, such as chlorogenic acid, caffeine, trigonelline, and magnesium, which are also associated with anti-obesity benefits [81].

### Limitations of the Study

This study has its limitations. The presented results are based on the questionnaire responses, so it shares all the limitations of this type of study. That is why the results should be taken with great caution. It was aimed to analyze the long-term dietary habits, so support by multiple biochemical tests regarding flavonol blood levels throughout a one-year period would be helpful. What is more, this study is also a retrospective model. A prospective and ideally interventional human study with biochemical support should confirm the presented results. The healthy controls were volunteers, so the majority of female participants was noted in the healthy control group; thus, the gender proportions in the healthy and central obese groups do not match the populational trends. The BIA parameters were also measured only in obese participants, so confirming the detailed relationships in healthy participants would be valuable. Nonetheless, this is the first study which analyzed the relationship between dietary flavonols as the single compounds and central obesity. This approach is highly valuable, as it provides a direction for future interventional studies regarding the preventive role of selected flavonols in obesity. This study was aimed to analyze general trends; however, a detailed investigation of the mechanisms responsible for these observations is needed, as it might be beyond the antioxidative properties.

## 5. Conclusions

Dietary flavonol intake could potentially present anti-obesity benefits. This study showed that participants with central obesity habitually consumed fewer flavonols (total flavonols, quercetin, kaempferol and isorhamnetin) than healthy participants. What is more, among the central obese participants, quercetin, kaempferol and myricetin intakes were inversely associated with parameters used for obesity assessment and classification, such as FM, waist circumference and BMI. Among major flavonol sources, tea presented strong correlation and coffee, moderate correlation, with obesity parameters in central obese participants. These results could suggest that a diet rich in flavonols, such as quercetin, kaempferol and isorhamnetin, with their main sources (tea and coffee), could be possibly protective against central obesity development.

## Figures and Tables

**Figure 1 nutrients-14-05051-f001:**
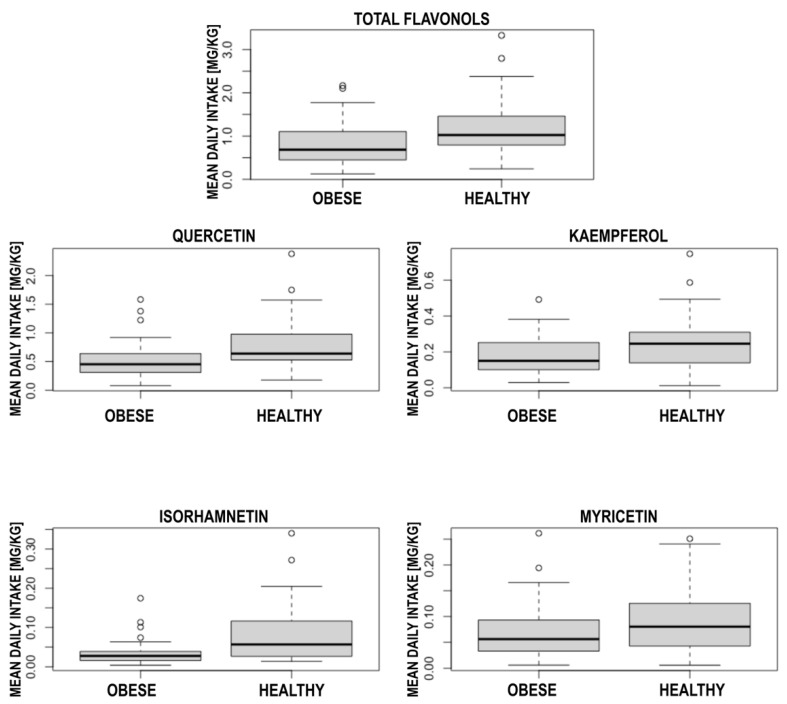
The boxplots presenting differences in flavonol intake between central obese participants and healthy control.

**Table 1 nutrients-14-05051-t001:** Anthropometrical characteristics of the central obesity group.

Anthropometrical Parameter [Unit]	Mean Value	SD
Body mass [kg]	80.18	±10.35
BMI [kg/m^2^]	29.18	±3.75
Waist circumference [cm]	102.72	±11.09
TBW [kg]	41.67	±6.00
TBW% [%]	52.26	±5.70
ECF [kg]	18.48	±2.95
ECF% [%]	44.39	±3.26
ICF [kg]	23.19	±3.71
ICF% [%]	55.63	±3.23
FFM [kg]	56.94	±8.17
FFM% [%]	71.36	±7.80
FM [kg]	23.12	±7.46
FM% [%]	28.61	±7.79

BMI—body mass index; TBW—total body water mass; TBW%—total body water percentage; ECF—extracellular fluid mass; ECF%—extracellular fluid percentage; ICF—intracellular fluid mass; ICF%—intracellular fluid percentage; FFM—fat-free mass; FFM%—fat-free mass percentage; FM—fat mass; FM%—fat mass percentage.

**Table 2 nutrients-14-05051-t002:** Comparison of the anthropometrical characteristics of the central obese and control group.

Anthropometrical Parameter [Unit]	Central Obesity (n = 39)	SD	Healthy Control (n = 40)	SD	*p*
Body mass [kg]	80.18	±10.35	59.48	±10.43	<0.001
BMI [kg/m^2^]	29.18	±3.75	20.94	±1.65	<0.001
Waist circumference [cm]	102.72	±11.09	65.6	±6.55	<0.001

BMI—body mass index.

**Table 3 nutrients-14-05051-t003:** The comparison of flavonol intake between central obesity participants and healthy control.

Mean Daily Intake [mg/kg]	Central Obesity (n = 39)	SD	Healthy Control (n = 40)	SD	*p*
Total flavonols	0.82	±0.44	1.18	±0.66	0.005
Quercetin	0.52	±0.29	0.77	±0.45	0.003
Kaempferol	0.19	±0.11	0.25	±0.15	0.04
Isorhamnetin	0.03	±0.02	0.08	±0.07	<0.001
Myricetin	0.07	±0.05	0.09	±0.06	0.19

**Table 4 nutrients-14-05051-t004:** Correlation between flavonol intake and BIA parameters, BMI and waist circumference.

Quercetin			
	R	95% CI	*p*
TBW	−0.127	−0.425; 0.197	0.44
TBW%	0.225	−0.097; 0.505	0.17
ECF	−0.131	−0.428; 0.193	0.43
ECF%	−0.011	−0.326; 0.305	0.95
ICF	−0.099	−0.402; 0.223	0.55
ICF%	0.004	−0.312; 0.319	0.98
FFM	−0.126	−0.424; 0.198	0.45
FFM%	0.215	−0.108; 0.497	0.19
FM	−0.352	−0.601; −0.041	0.03
FM%	−0.225	−0.505; 0.097	0.17
BMI	−0.336	−0.589; −0.023	0.04
Waist circumference	−0.322	−0.582; −0.002	0.05
Kaempferol			
	R	95% CI	*p*
TBW	−0.062	−0.371; 0.258	0.71
TBW%	0.330	0.016; 0.585	0.04
ECF	−0.083	−0.388; 0.239	0.62
ECF%	−0.024	−0.337; 0.294	0.89
ICF	−0.034	−0.345; 0.285	0.84
ICF%	0.021	−0.297; 0.334	0.90
FFM	−0.062	−0.370; 0.259	0.71
FFM%	0.321	0.006; 0.578	0.05
FM	−0.425	−0.653; −0.127	0.01
FM%	−0.330	−0.585; −0.016	0.04
BMI	−0.285	−0.551; 0.03	0.08
Waist circumference	−0.293	−0.560; 0.03	0.07
Isorhamnetin			
	R	95% CI	*p*
TBW	−0.034	−0.345; 0.285	0.84
TBW%	−0.029	−0.342; 0.289	0.86
ECF	0.013	−0.304; 0.327	0.94
ECF%	0.094	−0.229; 0.397	0.57
ICF	−0.064	−0.372; 0.257	0.70
ICF%	−0.102	−0.405; 0.220	0.54
FFM	−0.034	−0.346; 0.284	0.84
FFM%	−0.039	−0.350; 0.280	0.81
FM	0.003	−0.313; 0.318	0.99
FM%	0.029	−0.289; 0.342	0.86
BMI	−0.110	−0.411; 0.213	0.51
Waist circumference	−0.033	−0.349; 0.290	0.84
Myricetin			
	R	95% CI	*p*
TBW	−0.079	−0.385; 0.243	0.63
TBW%	0.254	−0.067; 0.527	0.12
ECF	−0.079	−0.385; 0.243	0.63
ECF%	0.011	−0.305; 0.325	0.95
ICF	−0.063	−0.371; 0.258	0.70
ICF%	−0.016	−0.330; 0.301	0.92
FFM	−0.078	−0.384; 0.244	0.64
FFM%	0.246	−0.076; 0.521	0.13
FM	−0.352	−0.601; −0.041	0.03
FM%	−0.254	−0.527; 0.067	0.12
BMI	−0.279	−0.546; 0.040	0.09
Waist circumference	−0.367	−0.615; −0.054	0.02
Total flavonols			
	R	95% CI	*p*
TBW	−0.109	−0.410; 0.214	0.51
TBW%	0.259	−0.062; 0.531	0.11
ECF	−0.114	−0.415; 0.209	0.49
ECF%	−0.007	−0.322; 0.309	0.97
ICF	−0.084	−0.389; 0.238	0.61
ICF%	0.0003	−0.315; 0.316	0.99
FFM	−0.108	−0.410; 0.215	0.51
FFM%	0.249	−0.073; 0.523	0.13
FM	−0.378	−0.620; −0.071	0.02
FM%	−0.259	−0.531; 0.061	0.11
BMI	−0.330	−0.584; −0.016	0.04
Waist circumference	−0.328	−0.586; −0.009	0.04

BMI—body mass index; TBW—total body water mass; TBW%—total body water percentage; ECF—extracellular fluid mass; ECF%—extracellular fluid percentage; ICF—intracellular fluid mass; ICF%—intracellular fluid percentage; FFM—fat-free mass; FFM%—fat-free mass percentage; FM—fat mass; FM%—fat mass percentage.

**Table 5 nutrients-14-05051-t005:** Correlation between flavonols’ sources mean consumption and obesity assessment parameters.

Fat Mass			
Product	R	95% CI	*p*
White onion	0.178	−0.146; 0.467	0.28
Red onion	−0.074	−0.381; 0.247	0.65
Tomatoes	0.283	−0.036; 0.549	0.08
Blueberry	0.043	−0.276; 0.354	0.79
Apple	−0.088	−0.393; 0.234	0.59
Black tea	−0.511	−0.712; −0.233	<0.001
Green tea	0.007	−0.310; 0.321	0.97
Coffee	−0.003	−0.318; 0.313	0.98
Wine	−0.100	−0.403; 0.223	0.55
**Fat mass %**			
Product	R	95% CI	*p*
White onion	0.145	−0.178; 0.441	0.38
Red onion	−0.120	−0.419; 0.204	0.47
Tomatoes	0.166	−0.158; 0.457	0.31
Blueberry	0.235	−0.087; 0.512	0.15
Apple	0.012	−0.305; 0.326	0.94
Black tea	−0.522	−0.719; −0.247	<0.001
Green tea	0.057	−0.263; 0.366	0.73
Coffee	0.183	−0.141; 0.471	0.26
Wine	−0.092	−0.396; 0.230	0.58
**Waist circumference**			
Product	R	95% CI	*p*
White onion	0.169	−0.159; 0.464	0.31
Red onion	0.031	−0.291; 0.347	0.86
Tomatoes	0.370	0.057; 0.617	0.02
Blueberry	−0.277	−0.548; 0.047	0.09
Apple	−0.018	−0.336; 0.303	0.91
Black tea	−0.201	−0.489; 0.127	0.23
Green tea	−0.141	−0.441; 0.187	0.40
Coffee	−0.352	−0.604; −0.036	0.03
Wine	−0.025	−0.342; 0.297	0.88
**BMI**			
Product	R	95% CI	*p*
White onion	−0.069	−0.377; 0.252	0.67
Red onion	−0.050	−0.360; 0.270	0.76
Tomatoes	0.283	−0.035; 0.550	0.08
Blueberry	−0.062	−0.370; 0.259	0.71
Apple	−0.052	−0.361; 0.268	0.76
Black tea	−0.311	−0.570; 0.005	0.05
Green tea	−0.010	−0.325 0.306	0.95
Coffee	−0.137	−0.434; 0.187	0.41
Wine	0.045	−0.275; 0.355	0.79

## Data Availability

The data that support the findings of this study are available from the corresponding author upon reasonable request.

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
