# Peer review of "The Impact of Dietary Flavonols on Central Obesity Parameters in Polish Adults"

_nutrients, 2022, doi:10.3390/nu14235051_

Round 1

Reviewer 1 Report

Manuscript Number: nutrients-2054884, titled:

 Dietary flavonols intake is associated with anti-obesity effects.

Review 1 – 15 November 2022

Dear Editor of Nutrients

1)    the argument is interesting but it has to be improved. The introduction section has to be extended with regards the flavonols contained in the species listed by authors. The M&M section has to be improved. The references section is not arranged as per Nutrients instructions for authors. Inaccuracies in the text.

I suggest a major revision

To the Authors (in detail):

2)    the argument is interesting but it has to be improved. The introduction section has to be extended with regards the flavonoids contained in the species listed by authors. The M&M section has to be improved. The references section is not arranged as per Nutrients instructions for authors. Inaccuracies in the text;

3)    Abstract section, you have indicated the symbol of the significance with p not italicized, whereas in many other parts of your manuscript you have italicized p. Please, be consistent in the whole manuscript: italicized or not?

4)    Abstract section: you have used the abbreviation CI, please, describe what means at the first mention;

5)    References section, separate the percentage value from the CI abbreviation, the same in the whole manuscript;

6)    Introduction section, lines 84-85, your work is related with dietary flavonols but you have described the flavonol content in foods, with only two lines and only 3 references. Please, extend, complete this sentence and improve your discussion. Include other proper and specific references related with foods such as: Onions [X1]; tea [X2]; apples [X3]; kale [X4]; lettuce [X5]; tomatoes [X6]; broccoli [X7]; grape [X8]; citrus fruits [X9, X10] (you have not listed them, even if they are very rich in flavonoids). Do not cumulate the references at the end of the sentence but insert the reference number after each related food. In particular, for grape, indicate in what part of the berry they are before to be transferred to wine. Extend the discussion with the influence of heat treatements and storage systems of foods in the flavonol content. Evidence the effect of cultivar on flavonols content. Support this discussion with proper references.

[X1] Onion (Allium cepa L.) bioactives: Chemistry, pharmaco therapeutic functions, and industrial applications

Food Frontiers.2022;3:380–412.

DOI:10.1002/fft2.135

[X2] Tea and flavonoids: where we are, where to go next

Am J Clin Nutr. 2013 Dec; 98(6): 1611S–1618S.

doi: 10.3945/ajcn.113.059584

[X3] Flavonoid and chlorogenic acid levels in apple fruit: characterisation of variation.

Scientia Horticulturae 83 (3–4) 2000, Pages 249-263

[X4] Characterization and quantification of flavonoids and hydroxycinnamic acids in curly kale (Brassica oleracea L. Convar. acephala Var. sabellica) by HPLC-DAD-ESI-MSn

J Agric Food Chem . 2009 Apr 8;57(7):2816-25. doi: 10.1021/jf803693t.

[X5] Lettuce flavonoids screening and phenotyping by chlorophyll fluorescence excitation ratio.

Planta, 2017 Jun;245(6):1215-1229. doi: 10.1007/s00425-017-2676-x.

[X6] The Flavonoids of Tomatoes.

J. Agric. Food Chem. 2008, 56, 7, 2436–2441

[X7] Effects of domestic cooking on flavonoids in broccoli and calculation of retention factors.

Heliyon. 2019 Mar; 5(3): e01310.

doi: 10.1016/j.heliyon.2019.e01310

[X8] HPLC-DAD detection of changes in phenol content of red berry skins during grape ripening.

European Food Research and Technology, 237 (4) 555-564 (2013)

DOI: 10.1007/s00217-013-2033-7

[X9] Citrus bergamia, Risso: the peel, the juice and the seed oil of the bergamot fruit of Reggio Calabria (South Italy).

Emirates Journal of Food and Agriculture 32(7) 522-532 (2020).

DOI: 10.9755/ejfa.2020.v32.i7.2128

[X10] Antioxidant Properties of Pulp, Peel and Seeds of Phlegrean Mandarin (Citrus reticulata Blanco) at Different Stages of Fruit Ripening

Antioxidants 2022, 11, 187. https://doi.org/10.3390/antiox11020187.

7)    sub-section 2.1, line 102: why the hyphen before 40 central?

8)    sub-section 2.1, please indicate if partecipants were smokers or not;

9)    sub-section 2.1, please, indicate the range of years of participants (³ 18 years is not enough);

10) sub-section 2.1, indicate the degree of instruction; of partecipants;

11) Tables 1,  right column, re-organize in relation with the ± symbol;

12) Caption of table 1, you have used different types of hyphens, please, uniform;

13) Caption of table 3, you have used different types of hyphens, please, uniform;

14) lines 265, 268 and in the whole manuscript, when you indicate a quantity, separate the numeric value from the symbol: 60 mg and not 60mg; 1000 mg and not 1000mg;

15) Line 310, delete one dot before … Numerous;

16) Line 311, delete one space after … the less;

17) Conclusions section, try to quantify and improve your conclusions with flavonols dietary intake;

18) The references section is not arranged as per Nutrients instructions for authors;

19) Please, write in blue color or evidence differently the corrections you will do.

I suggest a major revision

Regards.

Author Response

Dear Reviewer,

Thank you very much for your time, effort and valuable comments to my manuscript. I corrected the manuscript according to your suggestions and I believe that it deeply improved its quality. Below you can find my responses to your comments. Thank you for this opportunity.

To the Authors (in detail):

2)    the argument is interesting but it has to be improved. The introduction section has to be extended with regards the flavonoids contained in the species listed by authors. The M&M section has to be improved. The references section is not arranged as per Nutrients instructions for authors. Inaccuracies in the text;

The corrections were made according to the Reviewer suggestions. The introduction, material and methods, discussion sections were extended. The information about flavonols content in specific foods was added. The references list was prolonged as suggested by the Reviewer and arranged according to Nutrients guidelines.

3)    Abstract section, you have indicated the symbol of the significance with p not italicized, whereas in many other parts of your manuscript you have italicized p. Please, be consistent in the whole manuscript: italicized or not?

This aspect was corrected throughout the whole manuscript to make it consistent (italicized p).

4)    Abstract section: you have used the abbreviation CI, please, describe what means at the first mention;

This abbreviation was explained in the statistical analysis description (line 218).

5)    References section, separate the percentage value from the CI abbreviation, the same in the whole manuscript;

These corrections were made in the abstract section and throughout the rest of the manuscript.

6)    Introduction section, lines 84-85, your work is related with dietary flavonols but you have described the flavonol content in foods, with only two lines and only 3 references. Please, extend, complete this sentence and improve your discussion. Include other proper and specific references related with foods such as: Onions [X1]; tea [X2]; apples [X3]; kale [X4]; lettuce [X5]; tomatoes [X6]; broccoli [X7]; grape [X8]; citrus fruits [X9, X10] (you have not listed them, even if they are very rich in flavonoids). Do not cumulate the references at the end of the sentence but insert the reference number after each related food.  [X1] Onion (Allium cepa L.) bioactives: Chemistry, pharmaco therapeutic functions, and industrial applications

Food Frontiers.2022;3:380–412.

DOI:10.1002/fft2.135

[X2] Tea and flavonoids: where we are, where to go next

Am J Clin Nutr. 2013 Dec; 98(6): 1611S–1618S.

doi: 10.3945/ajcn.113.059584

[X3] Flavonoid and chlorogenic acid levels in apple fruit: characterisation of variation.

Scientia Horticulturae 83 (3–4) 2000, Pages 249-263

[X4] Characterization and quantification of flavonoids and hydroxycinnamic acids in curly kale (Brassica oleracea L. Convar. acephala Var. sabellica) by HPLC-DAD-ESI-MSn

J Agric Food Chem . 2009 Apr 8;57(7):2816-25. doi: 10.1021/jf803693t.

[X5] Lettuce flavonoids screening and phenotyping by chlorophyll fluorescence excitation ratio.

Planta, 2017 Jun;245(6):1215-1229. doi: 10.1007/s00425-017-2676-x.

[X6] The Flavonoids of Tomatoes.

  1. Agric. Food Chem. 2008, 56, 7, 2436–2441

[X7] Effects of domestic cooking on flavonoids in broccoli and calculation of retention factors.

Heliyon. 2019 Mar; 5(3): e01310.

doi: 10.1016/j.heliyon.2019.e01310

[X8] HPLC-DAD detection of changes in phenol content of red berry skins during grape ripening.

European Food Research and Technology, 237 (4) 555-564 (2013)

DOI: 10.1007/s00-7

[X9] Citrus bergamia, Risso: the peel, the juice and the seed oil of the bergamot fruit of Reggio Calabria (South Italy).

Emirates Journal of Food and Agriculture 32(7) 522-532 (2020).

DOI: 10.9755/ejfa.2020.v32.i7.2128

[X10] Antioxidant Properties of Pulp, Peel and Seeds of Phlegrean Mandarin (Citrus reticulata Blanco) at Different Stages of Fruit Ripening

Antioxidants 2022, 11, 187. https://doi.org/10.3390/antiox11020187.

The suggested citations were added in the proper spots (lines 113-117).  

In particular, for grape, indicate in what part of the berry they are before to be transferred to wine.

This information was added (line 116).

Extend the discussion with the influence of heat treatements and storage systems of foods in the flavonol content. Evidence the effect of cultivar on flavonols content. Support this discussion with proper references.

Thank you very much for this valuable remark. This information was added to discussion section with relevant references (lines 382-396).

7)    sub-section 2.1, line 102: why the hyphen before 40 central?

The information was taken in the brackets (line 139).

8)    sub-section 2.1, please indicate if partecipants were smokers or not;

This information was acknowledged in subsection 2.1 (lines 156-157).

9)    sub-section 2.1, please, indicate the range of years of participants (³ 18 years is not enough);

The range was given (lines 141-142).

10) sub-section 2.1, indicate the degree of instruction; of partecipants;

I am not sure if the Reviewer meant this sort of information, but the details of the questions asked to the respondents were given in the sections 2.1 and 2.2.1 (lines 143-144, 167-171).

11) Tables 1,  right column, re-organize in relation with the ± symbol;

Corrected.

12) Caption of table 1, you have used different types of hyphens, please, uniform;

Corrected.

13) Caption of table 3, you have used different types of hyphens, please, uniform;

Corrected.

14) lines 265, 268 and in the whole manuscript, when you indicate a quantity, separate the numeric value from the symbol: 60 mg and not 60mg; 1000 mg and not 1000mg;

Corrected.

15) Line 310, delete one dot before … Numerous;

Corrected.

16) Line 311, delete one space after … the less;

Corrected.

17) Conclusions section, try to quantify and improve your conclusions with flavonols dietary intake;

The conclusions section was developed (lines 500-509).

18) The references section is not arranged as per Nutrients instructions for authors;

The references were formatted according to Nutrients guidelines.

19) Please, write in blue color or evidence differently the corrections you will do.

The corrections were marked in blue color. All the changes are also highlighted with the “track the changes” option.

Reviewer 2 Report

Dear Editor of Nutrients

I have read with interest the manuscript proposed by Joanna Popiolek-Kalisz with title “Dietary flavonols intake is associated with anti-obesity effects” where he addresses such as the flavonols as antioxidative agents present in food may contribute to decrease the obesity central in overweight and obese patients. The proposal is interesting, however I have a series of questions that the manuscript addresses but that are not resolved satisfactorily.

Minor corrections

Page 1, line 2 please, change the title the manuscript or add preliminary study, because the patients were 40 and the population it is very low.

Page 1, line8, please, delete “by systemic low grade inflammation”….. and substitute by with systemic inflammation…..delete emerged as a risk factor… and substitute by drive to cardiovascular disease….

Page 1 line 40 and Page 8 line227 please delete the letter “s” in CVDs

Page 2, line 46 please change the phrase “…….expenditure leading to energy accumulation on the form of adipose tissue. And substitute by to increase the triglycerides which are stored in adipocytes

Page 2, line 57 and 75 please, add an abbreviation for insulin resistance (IR)

Page 2, line 70 please, add an abbreviation for fat mass (FM)

Page 2, line 73 please adds examples of inflammatory and anti-inflammatory cytokines

Page 2, line 93, please add the word Therefore after the “The aim of this paper….”

Page 2, line 98, please delete the phrase “This is the first paper which analized………” It is very risky to write this sentence since the number of patients is limited and because, in addition, through a questionnaire where the results are supported, it is insufficient and biochemistry cabinet studies are required to elucidate if this is correct.

Page 3 lines 145; please introduce a gap between ±SD, and this throughout the document and in the values of the tables.

Page 8 line 234 please deletes the phrase “This is the first paper which analized………” It is very risky to write this sentence since the number of patients is limited and because, in addition, through a questionnaire where the results are supported, it is insufficient and biochemistry cabinet studies are required to elucidate if this is correct.

Page 9 line 278 please add vs. because vs it is latin of the versus

Mayor corrections

Page 3 section 2.1. please the inclusion criteria are insufficient due to the hypothesis and proposed objective, the following criteria may modify the results body weight, height, alcohol consumption, narcotics, medicaments, daily exercise and duration and consumption of calories per day etc... Also please add the exclusion criteria for both groups of the patients control and obese

Please add the inclusion criteria, of the patients control for so that the comparison is equivalent

The study must be before and after in obese patients to show that the diet enriched with flavonoids showed a decrease in the degree of obesity.

Although the author mentions in the limitations of the study, that the results are based on a questionnaire on the consumption of foods that have a large amount of flavonoids, carried out on obese patients. However, the results should be taken with great caution, since there are no biochemical studies that demonstrate the decrease in pro inflammatory cytokines, the degree of obesity in patients.

Although the author mentions in the limitations of the study, that the populations was mayor in women in comparison with male, the study could be biased since obesity is higher in women than in men and there is not a good distribution

Please add in the tables the same results of the control subjects to be able to make an equivalent comparison and demonstrate what is intended in the objective of the study.

If possible support your results not only in a questionnaire since this is very subjective, and carry out biochemical studies of the blood chemistry cabinet.

Author Response

Dear Reviewer,

Thank you very much for your time, effort and valuable comments to my manuscript. I corrected the manuscript according to your suggestions and I believe that it deeply improved its quality. Below you can find my responses to your comments. Thank you for this opportunity.

Minor corrections

Page 1, line 2 please, change the title the manuscript or add preliminary study, because the patients were 40 and the population it is very low.

The title of the manuscript was changes as suggested by the Reviewer.

Page 1, line8, please, delete “by systemic low grade inflammation”….. and substitute by with systemic inflammation…..delete emerged as a risk factor… and substitute by drive to cardiovascular disease….

Corrected as suggested by the Reviewer (line 10).

Page 1 line 40 and Page 8 line227 please delete the letter “s” in CVDs

Corrected as suggested by the Reviewer (line 341).

Page 2, line 46 please change the phrase “…….expenditure leading to energy accumulation on the form of adipose tissue. And substitute by to increase the triglycerides which are stored in adipocytes

Corrected as suggested by the Reviewer (line 73-74).

Page 2, line 57 and 75 please, add an abbreviation for insulin resistance (IR)

Corrected as suggested by the Reviewer (line 85).

Page 2, line 70 please, add an abbreviation for fat mass (FM)

Corrected as suggested by the Reviewer (line 99).

Page 2, line 73 please adds examples of inflammatory and anti-inflammatory cytokines

Corrected as suggested by the Reviewer (lines 102-103).

Page 2, line 93, please add the word Therefore after the “The aim of this paper….”

Corrected as suggested by the Reviewer (line 125).

Page 2, line 98, please delete the phrase “This is the first paper which analized………” It is very risky to write this sentence since the number of patients is limited and because, in addition, through a questionnaire where the results are supported, it is insufficient and biochemistry cabinet studies are required to elucidate if this is correct.

Deleted as suggested by the Reviewer.

Page 3 lines 145; please introduce a gap between ±SD, and this throughout the document and in the values of the tables.

Corrected as suggested by the Reviewer.

Page 8 line 234 please deletes the phrase “This is the first paper which analized………” It is very risky to write this sentence since the number of patients is limited and because, in addition, through a questionnaire where the results are supported, it is insufficient and biochemistry cabinet studies are required to elucidate if this is correct.

Deleted as suggested by the Reviewer.

Page 9 line 278 please add vs. because vs it is latin of the versus

Corrected as suggested by the Reviewer.

Mayor corrections

Page 3 section 2.1. please the inclusion criteria are insufficient due to the hypothesis and proposed objective, the following criteria may modify the results body weight, height, alcohol consumption, narcotics, medicaments, daily exercise and duration and consumption of calories per day etc... Also please add the exclusion criteria for both groups of the patients control and obese

The inclusion and exclusion criteria were added in the section 2.1 as suggested by the Reviewer (lines 139-157).

Please add the inclusion criteria, of the patients control for so that the comparison is equivalent

The inclusion and exclusion criteria were added in the section 2.1 as suggested by the Reviewer (lines 139-157).

The study must be before and after in obese patients to show that the diet enriched with flavonoids showed a decrease in the degree of obesity.

This is a retrospective observational study and for this purpose the patients with central obesity were compared to healthy ones. As the study did not involve any intervention, it was not possible to present “before and after” data. The preliminary conclusions from this retrospective observation will definitely serve for future prospective and interventional studies upon this topic.

Although the author mentions in the limitations of the study, that the results are based on a questionnaire on the consumption of foods that have a large amount of flavonoids, carried out on obese patients. However, the results should be taken with great caution, since there are no biochemical studies that demonstrate the decrease in pro inflammatory cytokines, the degree of obesity in patients.

This is a very valuable note. That is why this note was underlined in the limitations of the study section. None the less the used questionnaire was proven valid for this sort of assessment.

Although the author mentions in the limitations of the study, that the populations was mayor in women in comparison with male, the study could be biased since obesity is higher in women than in men and there is not a good distribution

In Poland the central obesity prevalence is 45.7% for women and 32.2% for men in total population. When referred to specifically central obesity patients’ population the proportion is 60,2% women and 39,8% men among the central obese ones. Although the groups do not perfectly match this proportion it is underlined in the limitations of the study section. The information was also added in discussion (lines 336-338).

Please add in the tables the same results of the control subjects to be able to make an equivalent comparison and demonstrate what is intended in the objective of the study.

The suggested table was added (table 2), however the healthy participant were only screened using basic anthropometrical measurements, because the aim was to analyze the detailed trends in central obese patients. This information was also added to the limitations of the study section.  

If possible support your results not only in a questionnaire since this is very subjective, and carry out biochemical studies of the blood chemistry cabinet.

The study was planned as the retrospective, non-invasive type so the blood samples were not collected. What is more, the used questionnaire was proven suitable for this type of study in the following publication https://doi.org/10.3390/ijerph191912546

Round 2

Reviewer 1 Report

Manuscript Number: nutrients-2054884, titled:

 Dietary flavonols intake is associated with anti-obesity effects.

Review 2 – 23 November 2022

Dear Editor of Nutrients

1)    The Authors have included all my comments. The argument is interesting and well treated. The experiment design is well described and data are well discussed.

I suggest the publication of this manuscript in the present form.

Regards.

Reviewer 2 Report

Joanna Popiolek-Kalisk 

Thanks you for your responses, I dot not have more questions